# Declining grassland canopy height in China under asymmetric biomass allocation

Huaqiang Li ®[1,2,6], Xinmiao Hu[1,6], Fei Li ®[1,3,4] ✉, Yingjun Zhang ®[3] ✉, Kejian Lin[1], Jie Wang ®[3] & Jiating Wang[5]

Grassland canopy height is one of the most important traits for determining plant diversity and community structure, directly affecting the resource use efficiency of livestock in grassland ecosystems. However, broad-scale changes in grassland canopy height are seldom reported due to the complex effects of species aggregation on both interspecific and intraspecific structures. Here, we decouple grassland aboveground biomass into vertical and horizontal allocations, thereby offering a pathway to mirror changes in grassland canopy height. Grassland aboveground biomass is estimated using a machine learning algorithm by fusing climatic factors, satellite-driving metrics, and 8-year consecutive ground-truth surveys; the horizontal allocation of grassland aboveground biomass is derived from optimized linear spectral mixture analysis. We find that changes in horizontal biomass allocation primarily accounted for increases in Chinese grassland aboveground biomass from 2001 to 2022, resulting in a significant decline in grassland canopy height. The decline in grassland canopy height is shaped by reduced radiation and, more importantly, by the combined effects of warming and grazing, while also being related to variations in plant diversity. The dwarfing grassland community with declining canopy height may increase the impact of livestock disturbances, thus diminishing the resistance of grassland ecosystems to climate fluctuations.

Grassland canopy height is considered one of the most critical structural traits, serving as a fundamental indicator of functional diversity and resource use efficiency in grassland ecosystems[1]. At the functional trait level, grassland canopy height significantly influences plant ecological strategies, as taller species gain a competitive advantage in terms of light acquisition, potentially driving species turnover or loss, particularly under environmental pressures and intense grazing[2,3]. Therefore, grassland canopy height could serve as an underlying trait-based mechanism, reflecting the impact of manifold factors on morphological characteristics in different grassland species. A manipulative study on the Tibetan Plateau (2014–2017) found that rapid alpine warming shifted functional composition from legumes and forbs toward grasses and sedges, accompanied by an increase in canopy height[4]. By contrast, studies in arid regions report that warming shortens herbaceous growth cycles, limits biomass accumulation prior to seed set, and ultimately leads to vegetation dwarfing[5,6]. Grassland canopy height is also a key factor in explaining how livestock alter species diversity and community structure, with changes directly reflecting grazing intensity, influencing forage availability and quality, and consequently affecting the resource use efficiency of grassland ecosystems[7]. Therefore, accurately predicting grassland canopy height is crucial for understanding changes in ecosystem functionality and safeguarding food security in the future.

[1]Institute of Grassland Research, Chinese Academy of Agricultural Sciences, Hohhot, IM, China. [2]Tasmanian Institute of Agriculture, University of Tasmania, Launceston, Tasmania, Australia. [3]College of Grassland Science and Technology, China Agricultural University, Beijing, China. [4]Key Laboratory of Grassland and Agricultural Ecological Remote Sensing, Ministry of Agriculture and Rural Affairs, Hohhot, IM, China. [5]National Livestock Husbandry Station, Beijing, China. [6]These authors contributed equally: Huaqiang Li, Xinmiao Hu. ✉e-mail: lifei01@caas.cn; zhangyj@cau.edu.cn

Since the onset of the satellite era, optical remote sensing data have been widely used in the inversion-based upscaling of ecological metrics (e.g., biomass and cover)[8,9] owing to their high spatiotemporal resolution and provision of multidimensional spectral information. However, the effectiveness of optical remote sensing data for estimating canopy height is limited by signal saturation caused by a high level of fractional vegetation cover[1,10]. In domains with sparse vegetation cover (e.g., under 20%), soil noise interferes with the capture of plant signals, resulting in satellite metrics influenced by the interplay between plants and soils[11]. Although spaceborne light detection and ranging (LiDAR) instruments, such as the Global Ecosystem Dynamics Investigation (GEDI) mission, provide additional insights and techniques for estimating canopy structure, it is primarily focused on forests and has limited footprint coverage[12,13]. Compared with woody communities (e.g., forests and shrublands), grassland species are much smaller in size but exhibit complex interspecific and intraspecific structures, posing a challenge for directly extracting grassland canopy height with filtering algorithms because of the low signal-to-noise ratio of LiDAR data[14]. A study indicates that grassland canopy height models were sensitive to LiDAR point cloud density, as details of the maximum height of grassy canopies are often lost in cases with low-density point clouds[15]. Overall, existing reports on changes in grassland canopy height solely rely on plot-scale measurements from sparse experimental stations[4]; however, a robust strategy for tracking dynamics across broader spatial scales is lacking[16]. This knowledge gap limits our understanding of changes in grassland ecosystem functioning and resilience under climate extremes (e.g., droughts), thereby hindering effective grazing management and sustainable utilization.

Grassland species are primarily composed of herbaceous plants, which exhibit flexible allocational plasticity among their leaves, stems, and roots[17,18]. To maximize growth, plants in grasslands respond to environmental variations by partitioning biomass among different plant organs to optimize the capture of nutrients, light, water, and carbon dioxide[19,20]. Previous studies have shown that this allocation strategy is primarily manifested in biomass changes across horizontal and vertical dimensions, which explain a substantial variation in grassland aboveground biomass[4,21]. However, the direction (positive or negative) and intensity of biomass allocation across the vertical and horizontal dimensions remain controversial. Some studies of rapidly growing annual species have indicated that a competitive trade-off exists between the vertical and horizontal dimensions[20], where an increase in leaf area limits resource investment in roots and stems, resulting in shorter plant height[21]; however, others have argued that the plant community height in grasslands is positively correlated with leaf area[3,4]. These insights are valuable but are solely confined to the plot scale, creating a barrier to comprehensively understanding the principles of biomass allocation in grassland ecosystems at broader scales under the forcing of environmental change. In this regard, we hypothesize that the spatial allocation of grassland biomass differs between the vertical and horizontal dimensions due to environmental changes and grazing practices[22].

Our study is conducted across Chinese grasslands, and 24,125 field measurements of aboveground biomass, fractional vegetation cover, and canopy height were collected during the peak growing season (July–August) from 2009 to 2016 (Supplementary Fig. 1). We focused on China given its multiplex bioclimatic regimes and high stocking rate, which result in diverse species compositions and complex community structures; therefore, the canopy height estimation methods developed in this study can be applied elsewhere[8]. We propose that grassland aboveground biomass can be allocated across the horizontal fractional vegetation cover and vertical canopy height. Accordingly, the ratio of aboveground biomass to fractional vegetation cover ([aboveground biomass/ fractional vegetation cover]) reflects the vertical allocation of biomass per unit area and indicates variations in canopy height[1]. The gridded aboveground biomass at a 500 m resolution was simulated using a random forest machine learning algorithm with different assemblies of satellite observations, climate reanalysis data, and field-surveyed aboveground biomass; the horizontal determinant of the fractional vegetation cover was independently obtained using optimized linear spectral mixture analysis (SMA). Thereafter, we map the spatial regime of grassland canopy height and its changes during 2001–2022 using a strategy of [aboveground biomass/fractional vegetation cover] ratio. The main goal of this study is to investigate the spatial and temporal dynamics of grassland canopy height across China over the past 22 years and to elucidate the spatial allocation of aboveground biomass to broaden our understanding of shifts in ecosystem functioning and sustainability in the context of environmental change.

## Results and discussion
### Changes in grassland aboveground biomass
First, the random forest models of grassland aboveground biomass were trained using three forcing ensembles, namely, satellite-driving metrics, climatic factors, and their combination (Supplementary Table 1). The results revealed that the model with combined forcing factors achieved the highest accuracy ($R^2 = 0.73$) in comparison with the models driven by separate climatic factors ($R^2 = 0.69$) and satellite-driving metrics ($R^2 = 0.55$) (Fig. 1a). To further assess the consistency of the aboveground biomass variations from 2001 to 2022, we illustrated the interannual changes in aboveground biomass under different data forcings (Fig. 1b). The results revealed a consistently significant increasing trend, with rates ranging from 0.4 to 0.47 g m$^{-2}$ y$^{-1}$ and an ensemble average of 0.44 g m$^{-2}$ y$^{-1}$ ($t$ (20) = 3.48, $P = 2.4 \times 10^{-3}$, $R^2 = 0.38$, 95% CI [0.17, 0.70]). Moreover, we generated a trend map based on the ensemble average to examine the spatial changes in aboveground biomass (Fig. 1c). We observed that 63% of the grasslands in China exhibited an increasing aboveground biomass trend, with an average rate of 0.42 g m$^{-2}$ y$^{-1}$. Spatially, the aboveground biomass trend showed considerable heterogeneity, with a standard deviation (SD) of 1.25 g m$^{-2}$ y$^{-1}$. Significant increases in aboveground biomass were primarily identified in temperate montane meadow on the eastern Tibetan Plateau and in temperate meadow-steppe in eastern Inner Mongolia, with rates of 0.81 and 1.01 g m$^{-2}$ y$^{-1}$, respectively. In contrast, the alpine steppe in the central Tibetan Plateau showed a slight declining trend in aboveground biomass, with a rate of −0.05 g m$^{-2}$ y$^{-1}$. To further examine the robustness of aboveground biomass estimation, we calculated the SDs of the trends derived from the three forcing systems (Fig. 1d). The analysis revealed that 86% of the grassland aboveground biomass trends presented relatively low SDs (0–2 g m$^{-2}$ y$^{-1}$), with an average value of 1.23 ± 0.86 g m$^{-2}$ y$^{-1}$. We conclude that the grassland aboveground biomass in China has shown a robust increasing trend over the past 22 years, regardless of the forcing ensemble used.

### Changes in grassland fractional vegetation cover
Second, to reflect changes in horizontal aboveground biomass allocation, the fractional vegetation cover was calculated using SMA based on the satellite normalized difference vegetation index (NDVI). In particular, iterative optimization was employed to determine the endmembers of SMA (i.e., NDVI$_{veg}$ and NDVI$_{soil}$) across eight distinct grassland types (Supplementary Table 2). With these endmembers, the grassland fractional vegetation cover model exhibited strong performance (Supplementary Fig. 2a). In the horizontal dimension, aboveground biomass allocation significantly expanded from 2001 to 2022, with a trend of 0.24% y$^{-1}$ ($t$ (20) = 4.39, $P = 2.8 \times 10^{-4}$, $R^2 = 0.49$, 95% CI [0.001, 0.004], Fig. 2a). Spatially, horizontal aboveground biomass allocation exhibited a trend similar to that of the total aboveground biomass, and 70% of the grasslands experienced an increase from 2001 to 2022 (Fig. 2b). The significant increases were most evident in the temperate meadow-steppe and temperate steppe of Inner Mongolia,

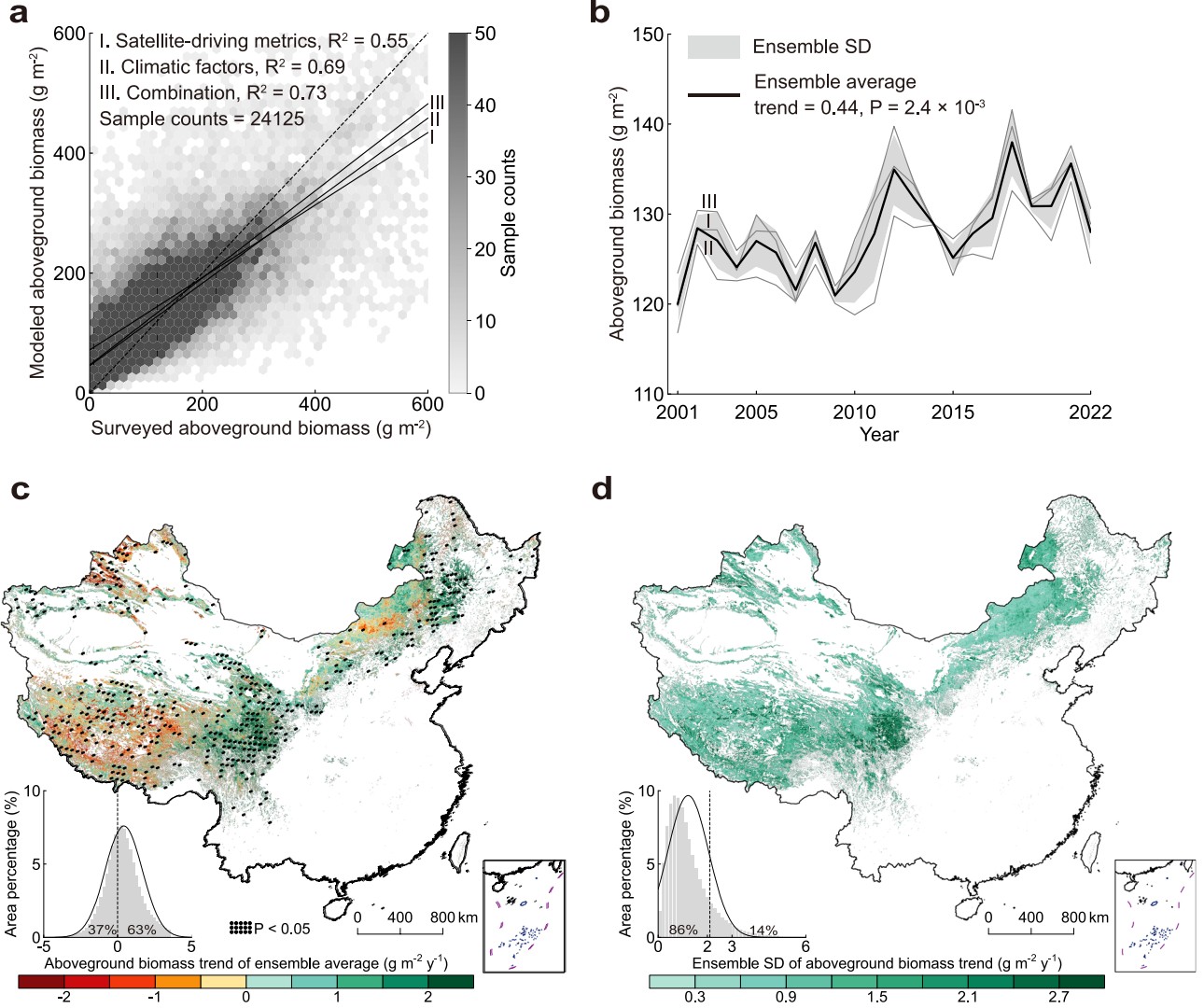

**Fig. 1 | Spatiotemporal changes in grassland aboveground biomass in China from 2001 to 2022 estimated using satellite-driving metrics and climatic factors separately and in combination. a** Holdout validation for random forest models with different forcing ensembles. **b** Interannual changes in grassland aboveground biomass. **c** Spatial regime of aboveground biomass trends, where trends are dotted if statistically significant (*P* < 0.05). **d** Spatial patterns of the ensemble SD of the aboveground biomass trends. The embedded histograms in the lower left of (**c**, **d**) are statistics of the area percentage aggregated by binning pixel values, with an overlaid normal curve for visualization. All the statistical significance in this study is assessed using a two-sided *t*-test, and no adjustment is made for multiple comparisons.

with rates of 0.44 and 0.46% y⁻¹, respectively. Notably, areas of decline were primarily observed in the central Tibetan Plateau, with a fractional vegetation cover trend of −0.17% y⁻¹. To further examine the contribution of horizontal aboveground biomass allocation to overall aboveground biomass changes, we calculated the Pearson correlation coefficient between the trends in fractional vegetation cover and aboveground biomass on a pixel basis (Fig. 2c). The results revealed a significantly positive correlation ($t$ (22079) = 67.5, $P < 1 \times 10^{-16}$, r = 0.41, 95% CI [109.19, 115.72]), with 70% of the grasslands showing a synchronous change. Overall, we found that the horizontal dimension of the aboveground biomass increased significantly and explained the aboveground biomass changes in 54% of the Chinese grasslands (spatial average $R^2$ = 0.26; Fig. 2d). However, the explanatory power in high fractional vegetation cover regions (e.g., the eastern Tibetan Plateau) was relatively low, accounting for less than 18%. We further found that when the fractional vegetation cover reached a plateau, the horizontally increasing capacity of the aboveground biomass was apparently constrained (Supplementary Fig. 3). Given the challenge of selecting strictly pure vegetation endmembers in sparse grasslands,

which may bias SMA-based fractional vegetation cover estimates, we instead used the classical gap fraction method to derive fractional vegetation cover, showing a similar result to that of the SMA fractional vegetation cover (Supplementary Note 1: Additional experiment on fractional vegetation cover calculation; Supplementary Figs. 2b, 4). We conclude that horizontal aboveground biomass allocation via fractional vegetation cover had a strongly synchronous response to the changes in the total grassland aboveground biomass from 2001 to 2022.

## Changes in grassland canopy height
Finally, grassland canopy height during 2001–2022 was estimated using a strategy involving the [aboveground biomass/fractional vegetation cover] ratio and linear regression analysis for scale convention in cm. On the basis of the order of the field-surveyed canopy height, we achieved an average accuracy of 73% in calculating grassland canopy height (Supplementary Fig. 5a). To further assess the accuracy of the modeled canopy height, we compared it with the surveyed canopy height across different grassland types, revealing a marginal difference

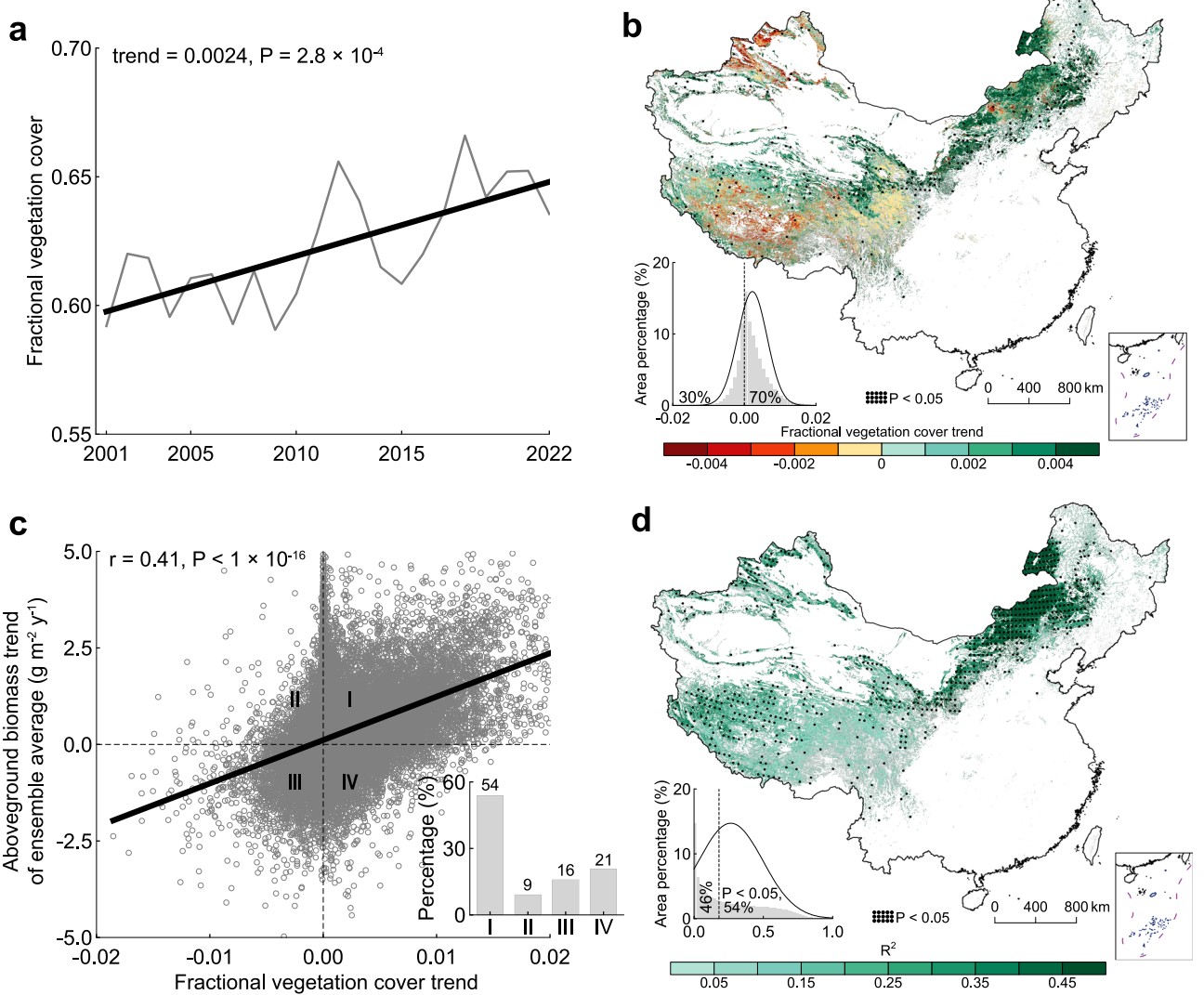

**Fig. 2 | Changes in fractional vegetation cover and the correlation with aboveground biomass across Chinese grasslands from 2001 to 2022. a** Interannual changes in fractional vegetation cover based on the SMA. **b** Spatial regime of fractional vegetation cover trends. **c** Correlations between changing rates in fractional vegetation cover and aboveground biomass on a pixel basis. **d** Spatial patterns of $R^2$ between changes in fractional vegetation cover and aboveground biomass, where $R^2$ are dotted if statistically significant ($P < 0.05$). The embedded histograms in (**b**–**d**) are statistics for the area percentage aggregated by binning pixel values, with an overlaid normal curve for visualization.

of 0.5 cm (left panel in Fig. 3a). The results showed that the grassland canopy height in China ranged from 0 to 40 cm, with an average height of $17.6 \pm 8.2$ cm (Fig. 3a). Spatially, grassland canopy height displays significant heterogeneity, decreasing from temperate grasslands ($27.4 \pm 8.5$ cm) in Inner Mongolia to alpine grasslands ($8.5 \pm 3.8$ cm) on the Tibetan Plateau. Among the seven grassland types, temperate meadow-steppe and lowland meadow in eastern Inner Mongolia have the highest canopy heights of 32.4 and 24.6 cm, respectively, whereas the alpine steppe and alpine desert-steppe in the western Tibetan Plateau presented the lowest heights ( ~ 8.0 cm). To further investigate the changes in grassland canopy height from 2001 to 2022, canopy height trends were calculated (Fig. 3b). We observed a noticeable decline in canopy height across 57% of the grasslands in China, with an average decreasing rate of $-0.04$ cm $y^{-1}$. These declining trends in canopy height across different grassland types exhibited notable heterogeneity, with an SD of 0.15 cm$^{-1}$ y$^{-1}$. In particular, canopy height in temperate and alpine types in northern and western China consistently decreased, ranging from $-0.21$ cm $y^{-1}$ in lowland meadow to $-0.01$ cm $y^{-1}$ in alpine steppe and alpine meadow (left panel in Fig. 3b).

Interestingly, canopy height in temperate meadow-steppe and temperate montane meadow (0.02 and 0.04 cm $y^{-1}$) in eastern Inner Mongolia and on the Tibetan Plateau slightly increased, indicating that the increase in aboveground biomass in high fractional vegetation cover areas was attributed primarily to the vertical dimension of biomass allocation. To further verify the robustness of the canopy height trend, we linearly fitted the [aboveground biomass/fractional vegetation cover] with different percentiles of field-surveyed canopy height, and the results were consistent (Supplementary Note 2: Additional experiment on canopy height trending robustness; Supplementary Figs. 5, 6). In summary, compared with the fractional vegetation cover increases in the horizontal dimension, our simulations reveal an asymmetric biomass allocation in the vertical dimension or a decrease in grassland canopy height across China during the contemporary period of 2001–2022.

## Independent evidence for declining canopy height
Previous studies indicate that $CO_2$ fertilization, together with climate warming-driven extensions of the growing season, has contributed to

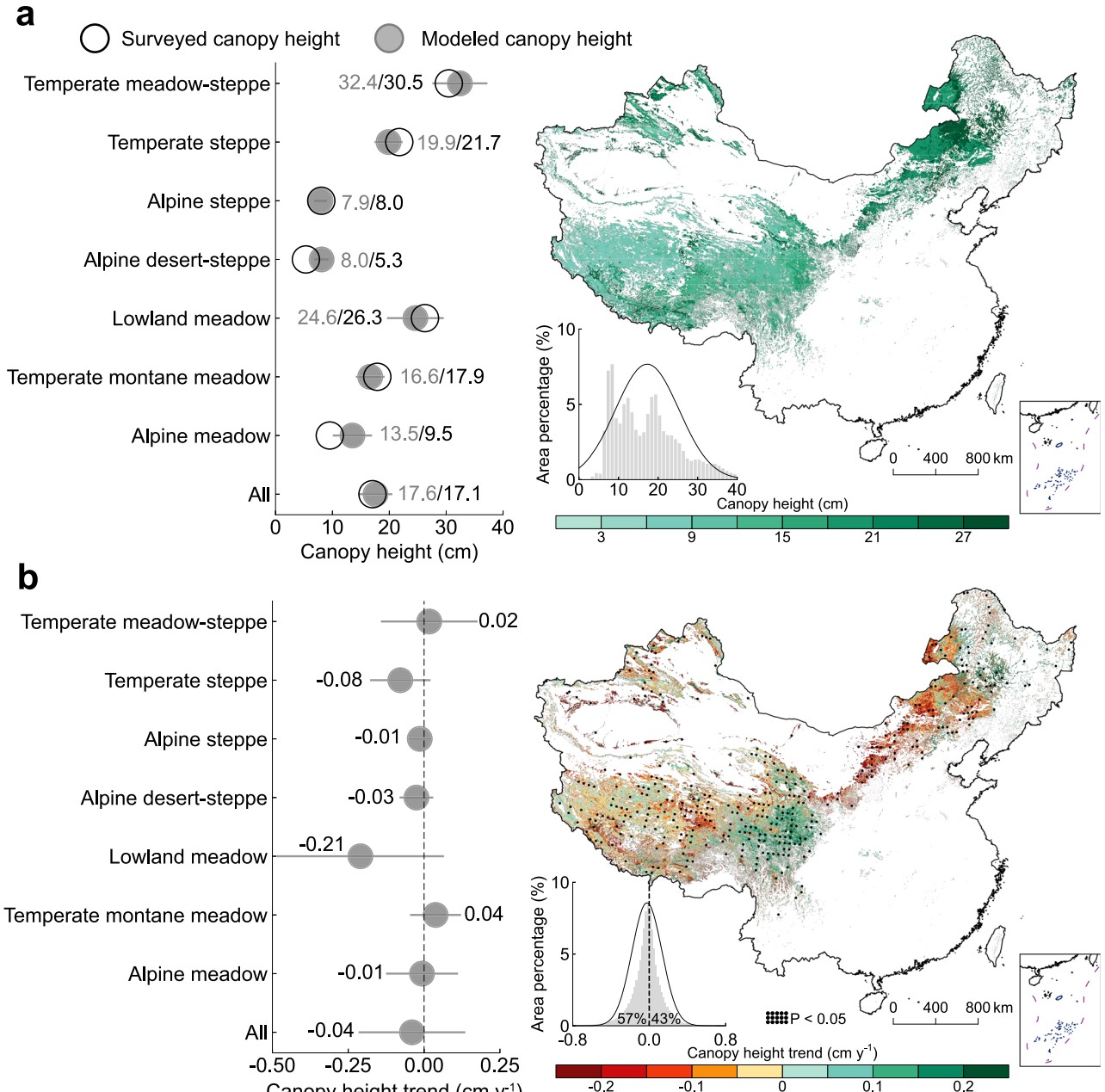

**Fig. 3 | Spatial regime of grassland canopy height and its trends across China from 2001 to 2022. a** Average grassland canopy height for seven grassland types (left panel) and spatial map (right panel). **b** Rates of grassland canopy height change for seven grassland types (left panel) and the trend map (right panel), where the trend is dotted if statistically significant (*P* < 0.05). The modeled canopy height and its trend in the left panels are presented as mean values ± SD. The embedded histograms in the right panels are statistics for the area percentage aggregated by binning pixel values, with an overlaid normal curve for visualization.

widespread terrestrial greening in recent decades[23], manifested by rising leaf area index (LAI) and concurrent increases in grassland biomass yield[24]. Our results indicate that horizontal aboveground biomass allocation seemingly accounts for a substantial proportion of biomass accumulation, resulting in a decrease in the vertical dimension of aboveground biomass allocation (i.e., canopy height). To further validate the trends of aboveground biomass changes in the vertical dimension, we collected continuous interannual aboveground biomass and fractional vegetation cover measurements (2001–2018) from six evenly distributed stations across China (Fig. 4). The direct measurements show a consistent declining trend in vertical aboveground biomass allocation during the contemporary period, with an average decreasing rate of −2.27 ± 1.09 g m$^{-2}$ y$^{-1}$. To account for the potential

limitation of using only the 2009–2016 survey years for temporal extrapolation, we assessed canopy height changes during this period and found that 53% of grasslands showed a decline, with an average trend of −0.02 cm y$^{-1}$ (Supplementary Fig. 7). For the more recent period of 2019–2022, GEDI-derived canopy height estimates also showed a decreasing trend (−0.09 m y$^{-1}$), but substantial uncertainty remains in characterizing grassland canopy height (Supplementary Fig. 8). Similarly, modeling uncertainties in aboveground biomass, fractional vegetation cover, and canopy height remain, particularly in the sparsely vegetated alpine desert steppe (Supplementary Note 3; Uncertainty analysis of canopy height estimation; Supplementary Fig. 9). Nevertheless, because canopy height trends are derived from interannual relative changes, their impact is limited to the magnitude

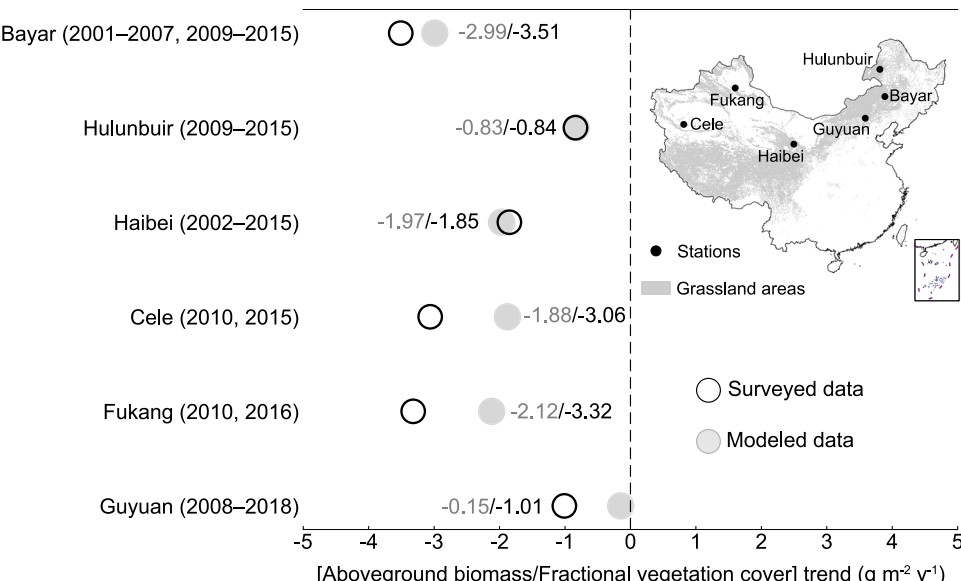

**Fig. 4 | Surveyed and modeled changes in the vertical allocation of aboveground biomass from the long-term grassland ecosystem observation stations across China.** The map in the right panel shows the spatial distribution of stations.

of the rate of canopy height decline rather than the direction of the trend, which remains robust overall.

The declining canopy height trend is also consistent with previous studies based on plot- or station-scale observations. For example, a significant reduction in grassland canopy height was reported under continuous grazing pressure, with a decrease approaching 38% (-−1.87 cm y[−1]) in three-year experiments (2009–2011) performed in the meadow steppe of Hulunbuir[7]. Similarly, a study conducted in two temperate grasslands in central and eastern Inner Mongolia indicated a decrease in community height between 2015 and 2018 due to drought, with decreases of −2.7 and −16.2 cm, respectively[25]. Globally, a meta-analysis of 197 studies reported a widespread shift in grassland species from tall (e.g., erect plants) to short plants (e.g., prostrate plants) under grazing impacts[26]. Overall, all the above evidence suggests a prevalent decline in grassland canopy height, whereas our study provides the dynamic and real-time evidence of this decline at a broad ecosystem scale.

**Driver analysis of asymmetric biomass allocation**

Grassland plant morphological characteristics (i.e., fractional vegetation cover and canopy height) are expected to undergo substantial changes under environmental change and grazing disturbances, but the underlying mechanisms remain unknown[4,27]. Here, we applied pixel-based structural equation modeling (SEM) with four environmental factors and grazing practice to evaluate the path effects of precipitation ($P_r$), air temperature ($T_a$), solar radiation ($R_a$), $CO_2$ fertilization effect ($C_a$), and grazing pressure ($G_p$) on canopy height and fractional vegetation cover. In areas with increased canopy height, the path coefficients indicated that simultaneous rises in $R_a$, $P_r$, and $C_a$ promoted the increases in canopy height and fractional vegetation cover, whereas $T_a$ and $G_p$ exerted negative effects on canopy height, with rising $T_a$ emerging as the dominant constraint (Fig. 5a).

Consistently, in areas with decreased canopy height, rising $P_r$ and $C_a$ continued to exert positive effects on both canopy height and fractional vegetation cover (Fig. 5b). However, increases in $T_a$ and $G_p$, together with reduced $R_a$, negatively influenced both canopy height and fractional vegetation cover, leading to an asymmetric spatial allocation of biomass, characterized by increasing fractional vegetation cover alongside decreasing canopy height. This shift reflects an adaptive trade-off in which plants tend to expand horizontally, as

vertical growth requires abundant rainfall and fertile soils, whereas horizontal expansion improves access to photosynthetic resources (radiation and $CO_2$)[2] and alleviates grazing damage[28]. Grazing manipulation experiments show that the combined effects of climate warming and grazing profoundly reshape grassland community composition and phenotypic traits, reduce plant diversity (e.g., species richness) and ecosystem multifunctionality, and ultimately lead to declines in biomass, canopy height, and community coverage[26,29]. Our results are consistent with previous studies, demonstrating that fractional vegetation cover, canopy height, and aboveground biomass are strongly influenced by climate warming and grazing disturbance, which jointly shape community structure and constrain canopy height development. Numerous studies indicate that rising $C_a$ and $P_r$ promoted fractional vegetation cover, canopy height, and biomass across different plant functional groups and species[30–33]. However, warming-induced water deficits can intensify grazing pressure, drive shifts in species composition[34], and ultimately suppress vegetation growth, manifested as grassland dwarfing and declines in canopy height[26].

We further investigated the interactive effects of climate warming and other driving factors on the spatial distribution of biomass using linear mixed-effects models (LMMs)[29]. The LMM results revealed consistently negative effects arising from increases in $T_a$ and $G_p$ (Supplementary Table 3). Moreover, we found a pervasive negative interaction between rising $T_a$ and other driving factors on both canopy height and fractional vegetation cover. In areas with decreased canopy height, the amplified impacts of the interaction between $T_a$ and $G_p$ were evident, exceeding their individual constraints on canopy height development. Collectively, we suggest that $CO_2$ fertilization, climate change, and grazing pressure jointly shape the spatial allocation of grassland biomass[29,31,35]. Among these factors, higher $R_a$, $P_r$, and $C_a$ promoted canopy height and fractional vegetation cover, whereas increases in $T_a$ and $G_p$, individually and in combination, exerted antagonistic effects[29,36], with canopy height being more strongly affected.

Previous studies have also shown that decreasing plant diversity weakens the stability and resistance of grassland ecosystems to environmental change[37,38]. Here, we examine the relationships between surveyed species richness, GEDI-derived foliage height diversity (FHD)[13], and canopy height trends. Our results indicate that canopy height trends respond linearly and significantly to species

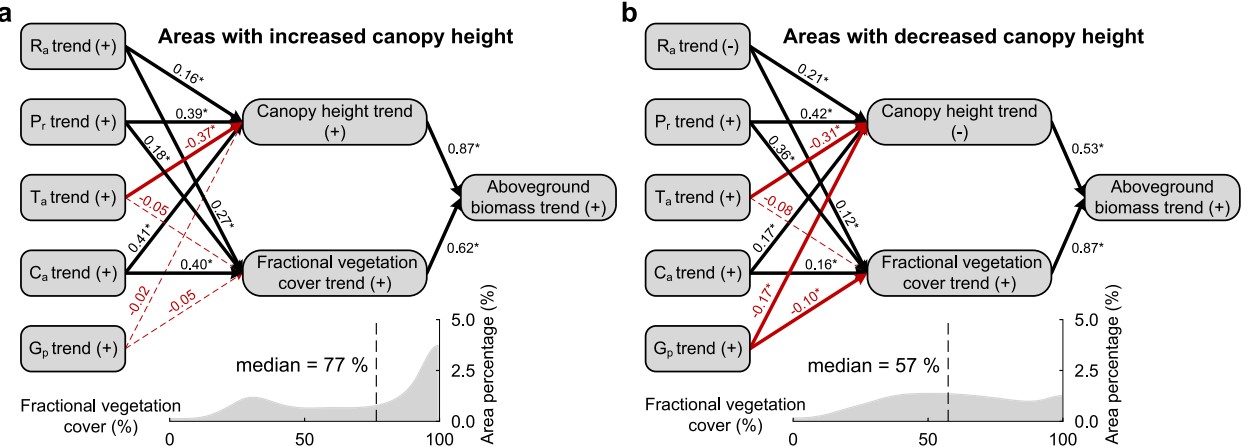

**Fig. 5 | Responses of canopy height and fractional vegetation cover to environmental changes in solar radiation ($R_a$), precipitation ($P_r$), air temperature ($T_a$), $CO_2$ fertilization effect ($C_a$), and grazing pressure ($G_p$). a, b** Path coefficients for grassland areas with increased canopy height (**a**), and decreased canopy height (**b**), with the calculation of SEM, where the coefficients are asterisked (*) if statistically significant ($P < 0.05$). The positive (+) and negative (−) signs indicate the direction of the corresponding trend. The embedded area charts present the overall and median statistics of the 2001–2022 average fractional vegetation cover for these two contrasting regions.

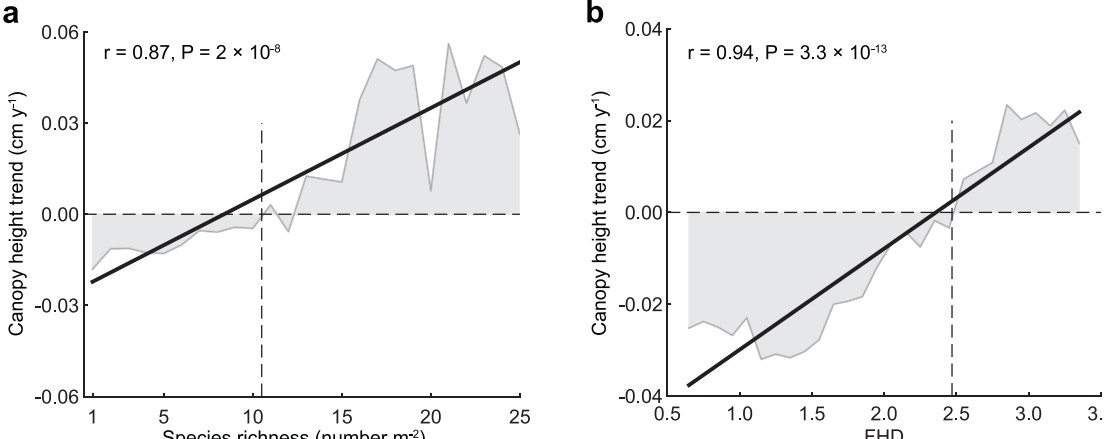

**Fig. 6 | Correlation between 2001–2022 canopy height trends and plant diversity across Chinese grasslands. a** Species richness. **b** GEDI-derived foliage height diversity (FHD).

richness and FHD, with correlation coefficients of 0.87 ($t$ (23) = 8.35, $P = 2 \times 10^{-8}$, 95% CI [0.002, 0.004]) and 0.94 ($t$ (26) = 13.44, $P = 3.3 \times 10^{-13}$, 95% CI [0.019, 0.025]), respectively (Fig. 6). Meanwhile, a tipping point was observed whereby species richness below 11 showing an apparent decline in canopy height (Fig. 6a). Similarly, grassland canopy height with FHD below 2.47 exhibited a negative trend (Fig. 6b). These findings suggest grazing-induced species loss can disrupt asynchronous responses among species to climate variability, thereby intensifying canopy height reductions, particularly during drought years[22,39].

In conclusion, the grassland biome in China, as an important part of the Eurasian steppes, plays a crucial role in maintaining ecological services, ensuring food security, and promoting sustainable development, serving as an important component for achieving the global Sustainable Development Goals (SDGs)[40]. Previous studies have consistently reported that grassland productivity and greenness have increased over the past few decades[23], seemingly illustrating a positive signal of ecosystem evolution in the context of climate warming[24,41]. Unfortunately, we observed a significant decline in grassland canopy height from 2001 to 2022, reflecting an emerging alteration in plant

morphological characteristics. The decrease in grassland canopy height will not only cause ruminants (e.g., cattle and sheep) to increase the difficulty and grazing time for forage accessibility, but also enhance the risk of disturbance to grassland roots during grazing[26]. Moreover, decreased grassland canopy height can further reduce ecosystem structural diversity and functional sustainability, resulting in strong productivity fluctuations in response to climate extremes and human activities[42]. Our findings reveal that grassland canopy height decline is notably shaped by ongoing climate warming and grazing pressure, and is closely associated with shifts in species diversity. We advocate climate-adaptive management that pairs rotational and rest grazing with revegetation to sustain species richness and structural diversity, thereby enhancing ecosystem resistance and resilience to achieve the SDGs.

## Methods
### Datasets
Optical remote sensing images were obtained from the Terra Moderate Resolution Imaging Spectroradiometer (MODIS), which is operated by the National Aeronautics and Space Administration (NASA), as part

of the Earth Observing System (https://modis.gsfc.nasa.gov). We used two products, MOD09A1 and MOD15A2H, covering the peak growing season (July–August) from 2001 to 2022. MOD09A1 was used to calculate 11 satellite-driving metrics (see Supplementary Table 1). The spectral bands used included seven bands at a 500 m resolution: blue (469 nm), green (555 nm), red (645 nm), near-infrared (859 nm), shortwave infrared 1 (1240 nm), shortwave infrared 2 (1640 nm), and shortwave infrared 3 (2130 nm). We used MOD15A2H LAI data to calculate fractional vegetation cover using the classical gap fraction method from 2001 to 2022. GEDI Level 2 A and Level 2B products were employed to extract relative height metrics and FHD[13], respectively, from 2019 to 2022. All the remote sensing data were processed using maximum value compositing for July and August via the Google Earth Engine (GEE) platform.

Climate data were derived from the ERA5-Land dataset provided by the Copernicus Climate Change Service through the European Center for Medium-Range Weather Forecasts (https://cds.climate.copernicus.eu/). This dataset provides monthly meteorological data on a 0.1° (~10 km) latitude/longitude grid. We utilized monthly averages for $P_r$, $T_a$, and $R_a$ to calculate the 22-year mean annual precipitation, mean annual temperature, mean annual vapor pressure deficit, and mean annual $R_a$, as well as interannual and seasonal anomalies (June–September) relative to the 22-year averages from 2001 to 2022 (see Supplementary Table 1). Additionally, we applied a linear correction to ERA5 precipitation data via station data to eliminate systematic errors. The station data of monthly precipitation from 2001 to 2019 were obtained from 134 long-term monitoring meteorological stations managed by the China Meteorological Data Service Center (https://data.cma.cn/en). The monthly averaged gridded $C_a$ data at 0.75° spatial resolution from 2003 to 2020 were obtained from the Copernicus Atmosphere Monitoring Service (CAMS) global greenhouse gas reanalysis[43].

The measurements of aboveground biomass, fractional vegetation cover, canopy height and species richness were derived from the National Inventory of Grassland Resources by the National Forestry and Grassland Administration of China from 2009 to 2016 (http://www.forestry.gov.cn), and 24,125 field plots were collected during the peak growing season (July–August) (Supplementary Fig. 1). The aboveground biomass was measured by harvesting the aboveground portion of the plants and oven-drying them to obtain the dry mass weight. The fractional vegetation cover was measured using the point quadrat method, in which a grid with needles was placed over the plot, and the ratio of needle hits on vegetation versus soil was recorded to estimate cover. Canopy height was measured at the center of the plot with a ruler, and species richness was assessed through field quadrat surveys. The grazing intensity data were collected from ref. 44., with a spatial resolution of ~250 m, covering the period from 2001 to 2022.

The shapefile data for grassland types at a 1:1 million scale were utilized, spanning temperate meadow-steppe, temperate steppe, temperate desert-steppe, alpine steppe, alpine desert-steppe, lowland meadow, temperate montane meadow, and alpine meadow. The data for continuous interannual measurements of aboveground biomass and fractional vegetation cover at six observation stations were primarily sourced from the China Ecosystem Positioning Observation and Research Dataset (www.nesdc.org.cn), with additional data from ref. 45.

### Aboveground biomass modeling

Grassland aboveground biomass was estimated through random forest models driven by per se satellite-driving metrics, climatic factors, and their combination, respectively (Supplementary Table 1). The holdout and five-fold cross-validation schemes were employed for model training. First, the total number of training samples was randomly divided, with 10% allocated for independent validation and the remaining 90% used for iterative training with five-fold cross-

validation. In each iteration, four subsets of samples were sequentially selected for model training, and the single remaining subset was used for model cross-validation[8]. The training was completed four times to calculate the average validation error across the holdout and five-fold cross-validation datasets. To improve the underestimation of high aboveground biomass values during model training, the random oversampling method was employed to balance the training samples by increasing the number of minority data instances (i.e., samples with high aboveground biomass values)[46]. Moreover, to avoid excessive dependence on individual trees, Bayesian model averaging[47] was applied to calculate the results of the random forest model. Given the availability of ground-truth data, the final aboveground biomass upscaling models with three forcing ensembles were established using the full dataset through a five-fold cross-training scheme. Using the ensemble average of the aboveground biomass models, we generated gridded aboveground biomass for Chinese grasslands at a spatial resolution of 500 m and yearly time steps from 2001 to 2022, which were then used to calculate interannual changes, spatial trends, and the SD map reflecting different forcing ensembles. The above calculation was performed using the RandomForestRegressor module from scikit-learn 1.6.1 and the RandomOverSampler module from imbalanced-learn 0.13.0 in Python 3.10.16.

### Fractional vegetation cover calculation

The grassland fractional vegetation cover was calculated for each of the eight grassland types using the SMA method[9]. During the SMA process, based on the survey data, an iterative optimization method was employed to determine the optimal endmembers (i.e., $NDVI_{soil}$ and $NDVI_{veg}$) for each grassland type, as detailed in Supplementary Table 2. With these optimal endmembers and NDVI composite values, fractional vegetation cover was calculated on the GEE platform using JavaScript with the normalizedDifference function.

### Canopy height calculation

Grassland canopy height (cm) was calculated using a linear model on the basis of the relationship between the surveyed canopy height and the ratio of aboveground biomass to fractional vegetation cover (Supplementary Fig. 5a). The linear quantile regression of the surveyed canopy height with the [aboveground biomass/fractional vegetation cover] ratio was determined to control the systematic errors in canopy height estimation across the eight grassland types. With the linear model for each grassland type, we converted gridded [aboveground biomass/fractional vegetation cover] ratios to canopy heights at a spatial resolution of 500 m and yearly time steps from 2001 to 2022. To further assess the robustness of the trends in the canopy height calculation, the linear model was tested with different statistical schemes (e.g., average), as detailed in Supplementary Note 2: Additional experiment on canopy height trending robustness, and Supplementary Figs. 5b, 6.

### The driving factor analysis of canopy height and fractional vegetation cover

The piecewise SEMs[48] were applied to evaluate the effects (i.e., path coefficients) of environmental factors and grazing intensity on canopy height and fractional vegetation cover in relation to changes in aboveground biomass. First, pixel-based trends of explanatory factors ($R_a$, $P_r$, $T_a$, $C_a$, and $G_p$) and response variables (canopy height, fractional vegetation cover, and aboveground biomass) were calculated for the period 2001–2022. For cross-comparison of the effects, explanatory factors were normalized using the z-score method by subtracting the mean of each term and dividing by the standard deviation. Second, pixel-scale SEMs were constructed for Chinese grasslands classified by increasing or decreasing canopy height, using normalized explanatory and response trends, with sample sizes of 4.7 and 6.3 million, respectively. In addition, we incorporated trends of explanatory factors as

pairwise combinations (e.g., $\Delta T_a \times \Delta G_p$), and applied Ms to examine both their independent and interactive effects on the response variables (Supplementary Table 3). The above analyses were performed using R statistical software version 4.5.1. The SEMs were carried out using the psem function of the piecewiseSEM package, and LMMs were performed using the lme function within the nlme package.

## Reporting summary
Further information on research design is available in the Nature Portfolio Reporting Summary linked to this article.

## Data availability
The MOD09A1 data can be accessed at https://doi.org/10.5067/MODIS/MOD09A1.061, and the MOD15A2H data from https://doi.org/10.5067/MODIS/MOD15A2H.061. GEDI Level 2 A and Level 2B products are available at https://doi.org/10.5067/GEDI/GEDI02_A.002 and https://doi.org/10.5067/GEDI/GEDI02_B.002, respectively. The ERA5-Land dataset can be downloaded at https://cds.climate.copernicus.eu/. The $CO_2$ data can be accessed at https://doi.org/10.24381/a90c7e33. The field measurements of aboveground biomass, fractional vegetation cover, canopy height, and species richness are collected from the National Inventory of Grassland Resources by the National Forestry and Grassland Administration of China. The grazing intensity data are available at https://doi.org/10.6084/m9.figshare.26195684. The data for continuous interannual measurements of aboveground biomass and fractional vegetation cover are primarily obtained from the China Ecosystem Research Network (www.nesdc.org.cn). The gridded dataset of grassland canopy height across China (2001–2022) generated in this study is available in the Zenodo repository [https://doi.org/10.5281/zenodo.18454934][49].

## Code availability
The codes in this study are available in the Zenodo repository [https://doi.org/10.5281/zenodo.18453938][50].

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

## Acknowledgments

This project was supported by the National Natural Science Foundation of China (42471426). F.L. was supported by the Chinese Universities Scientific Fund (2025TC044) and the Science and Technology Program of Inner Mongolia Autonomous Region (2025YFDZ0055).

## Author contributions

H.L. conducted model simulations, analysed the results, and drafted the text. X.H. conducted driver analysis and drafted the text. F.L. designed the study, analysed the results, and edited the text. Y.Z. contributed to the improvement of the study design and the drafting of the text. K.L. and J.W. (Jie Wang) revised the manuscript. J.W. (Jiating Wang) contributed to the collection of ground-truth data.

## Competing interests

The authors declare no competing interests.
