## [Peer Review file · Nature Communications]

Declining grassland canopy height in China under asymmetric biomass allocation

Corresponding Author: Dr Fei Li

Version 0:

Reviewer comments:

Reviewer #1

(Remarks to the Author)

Your work shows a detailed and interesting study on a very important topic currently. I am sending some questions and suggestions to clarify some points that might be better explained and contribute even more to the subject.

1. Regarding the estimate of CH from the relation/equation AGB/FVC: please provide reference and example of its use elsewhere.
2. Line 88: please add "respectively" for FVC and CH.
3. How did you manage the different range of field data (2009 to 2016) and period of study (2001 to 2022)? In addition, in which context data from the six stations of NESDC were used?
4. Information about LAI was measured also in the field? Please explain further the use of MOD15A2H 289 LAI data to calculate FVC using the classical gap fraction method.
5. The collection of pure pixels was made for the green pixels, and the soil pixels were derived from them? It is difficult to imagine pure pixels for a 500 m spatial resolution.
6. It is not clear why allocation of light varied or have an impact on CH under fertilizing effect of rising CO₂.
7. It would be important to add a comment on the effect of diminishing CH on the diversity of grass species.

Reviewer #2

(Remarks to the Author)

The study reveals China's grassland canopy height (CH) is declining, affecting ecosystem function and food security. It highlights vulnerability to climate change and grazing, guiding adaptive management strategies.

The AGB/FVC ratio method for estimating CH in grasslands addresses remote sensing challenges due to low-density point clouds in LiDAR data, integrating machine learning with multi-source data for high-resolution monitoring.

This study uses the AGB/FVC ratio to estimate grassland CH at a national scale, addressing a gap in previous studies. It provides broad-scale evidence of CH decline, offering a dynamic perspective not previously captured in plot-scale studies.

The work is unique in its methodology for CH estimation and comprehensive analysis of China's grasslands over two decades, building on established remote sensing and ecological principles.

+ The study's reliance on field data from 2009–2016, which does not span the full analysis period (2001–2022), may introduce bias in trend extrapolation, and the use of 500 m resolution data may miss fine-scale variations in CH. To address these limitations, it is recommended to extend the field dataset by including earlier or more recent measurements or validate findings using independent datasets. Using higher-quality satellite data like Sentinel-2 can provide more detailed information, and carefully adding GEDI LiDAR data can help confirm canopy height measurements, even though there are some issues with grasslands. A dedicated section or supplementary analysis should be included to quantify the impacts of these temporal and spatial constraints.

+ The manuscript does not provide a detailed analysis of the uncertainties in CH estimates, especially in places with low FVC, where optical remote sensing can give misleading results. To make this better, the authors should analyze how errors affect the CH estimates that come from the AGB/FVC ratio, taking into account the uncertainties in both the AGB and FVC models. Uncertainty should also be explicitly assessed across FVC gradients, particularly in sparse vegetation regions, and confidence intervals should be reported. Additionally, cross-validation metrics such as RMSE and bias should be expanded across grassland types. A dedicated subsection with visualizations like uncertainty maps is recommended to present these results clearly.

+ The paper's claims about reduced ecosystem resilience, biodiversity loss, and grazing impacts are plausible but lack direct supporting evidence, particularly given the non-significant grazing variable. To make these points stronger, the authors should include or reference data on biodiversity, functional traits, or soil carbon to show how the CH decline is connected to ecosystem damage. More detailed grazing data or sensitivity analyses could clarify grazing's role. The authors are also encouraged to propose actionable management strategies, such as rotational grazing or restoration of tall-grass species—and relate these to relevant Sustainable Development Goals.

+ The current explanation of asymmetric biomass allocation in the study is overly simplified, emphasizing light optimization without addressing underlying physiological or ecological mechanisms. To deepen the analysis, the authors should integrate data or models on traits like leaf area index (LAI), root biomass, or photosynthetic efficiency to explain growth patterns, and consider interaction effects among drivers such as CO₂ and precipitation.

+ The paper should improve its global relevance by integrating its findings on China's declining grassland canopy height with global trends. It should compare China's CH decline with other grassland ecosystems, using references like Sloat et al. (2018) and Díaz et al. (2007). Also, highlight unique factors like intensive grazing or climate variability.

Version 1:

Reviewer comments:

Reviewer #1

(Remarks to the Author)

I acknowledge and thank you for the careful review you made to the manuscript. The questions and suggestions I made were sorted and helped to make the manuscript more relevant and understandable. Also, the reproductive potential of the steps you took were also assured, which make an article effective and useful. As it addresses important questions regarding climate change, it is vital to make it accessible to a broader audience that can make a difference for our future.

(Remarks on code availability)

made.

Response to Reviewers' Comments

We thank the reviewers for the positive reviews and constructive comments on our manuscript. In preparing this revision, we have fully considered the reviewers' comments, conducted additional experiments and analyses as suggested, and revised the manuscript accordingly. We summarize our responses to these comments below in blue text, with reference to additional response figures and tables (Figs. R1 to R8, Tables R1 and R2). All line numbers refer to numbers in the revised manuscript unless noted otherwise.

Reviewer #1 (Remarks to the Author):

Your work shows a detailed and interesting study on a very important topic currently. I am sending some questions and suggestions to clarify some points that might be better explained and contribute even more to the subject.

1. Regarding the estimate of CH from the relation/equation AGB/FVC: please provide reference and example of its use elsewhere.

Response: Thanks for the suggestion. We revised the manuscript to further clarify the definition of AGB/FVC, and explained why it can be used to derive CH, with supporting evidence from the references. First, we clarified that AGB/FVC represents the biomass per unit area, which we use as an indicator of the vertical allocation of AGB, rather than CH itself. Many studies have demonstrated a strong linear positive correlation between aboveground biomass and plant height (Lang et al., 2023; Proulx, 2021). These evidences provide the basis and likelihood for estimating CH using vertical biomass and linear regression. Our results also show a significantly linear relationship between the ratios of AGB/FVC and surveyed CH across different grassland types (Supplementary Fig. 5). These results further demonstrate the rationality of estimating CH using AGB/FVC. We have modified the main document to clarify the definition and application of AGB/FVC (lines 93–94).

References:

1. Lang, Nico, et al. A high-resolution canopy height model of the Earth. *Nature Ecology & Evolution* 7.11 (2023): 1778-1789.
2. Proulx, Raphaël. On the general relationship between plant height and aboveground biomass of vegetation stands in contrasted ecosystems. *PLoS One* 16.5 (2021): e0252080.

2. Line 88: please add “respectively” for FVC and CH.

Response: We restructured this sentence in the main text to avoid semantic ambiguity (lines 91–92).

3. How did you manage the different range of field data (2009 to 2016) and period of study (2001 to 2022)? In addition, in which context data from the six stations of NESDC were used?

Response: Thank you for raising this critical issue. Initially, the surveyed samples of AGB, CH, and FVC from 2009–2016 were modeled with the expectation of reflecting long-term changes over 2001–2022. However, this inevitably raises the question of how to validate changes outside the surveyed window, both prior to 2009 and after 2016.

To address this concern, we conducted segmented experiments, as direct field measurements covering the entire study period are almost unavailable. For the early period (2001–2018), we collected vertical biomass data from six long-term NESDC observation sites to corroborate the declining trend during gaps in the early years (Fig. R1). We also compared the observed decline with model-extrapolated results, and found consistent trends, despite some biases at two of the six sites caused by scale effects and station continuity.

For the middle period (2009–2016), where the models were directly constrained by field data, we derived CH trends from simulated gridded data inferred from AGB/FVC. The results confirmed a consistent decreasing trend, with an average rate of -0.02 cm yr^{-1} , and $\sim 53\%$ of China's grasslands exhibited decline (Fig. R2).

For the late period (2019–2022), we used GEDI Level-2A data to extract relative canopy height metrics. GEDI-derived CH also showed an overall decline of -0.09 m yr^{-1} , with consistent decreases detected across five grassland types (Fig. R3).

In summary, although the magnitude of decline varied slightly among datasets, all independent CH estimates consistently revealed a robust decreasing trend. To highlight this convergence of evidence, we have revised the main document (lines 209–218), added Fig. R1 (renumbered as Fig. 4), and updated the Supplementary Material with Fig. R2 and Fig. R3 (now Supplementary Figs. 7 and 8).

Fig. R1. Surveyed and modelled changes in the vertical allocation of AGB from the long-term grassland ecosystem observation stations across China. The map in the right panel shows the spatial distribution of stations.

Fig. R2. Spatiotemporal changes in grassland CH in China from 2009 to 2016. a, Interannual changes. b, Rates of grassland CH change for seven grassland types (left panel) and the trend map (right panel), where the trend is dotted if statistically significant ($P < 0.05$); the embedded histograms in the right panels are statistics for the area percentage aggregated by binning pixel values.

Fig. R3. Changes of CH derived from the GEDI dataset across Chinese grasslands from 2019 to 2022. a, Interannual changes of CH. b, Rates of CH change for seven grassland types.

4. Information about LAI was measured also in the field? Please explain further the use of MOD15A2H 289 LAI data to calculate FVC using the classical gap fraction method.

Response: As noted, the SMA approach is one of the most widely used methods for estimating FVC; however, selecting strictly pure endmembers can be challenging in sparse grasslands. To overcome this limitation, we also applied the classical gap fraction method, which does not rely on endmember selection but instead requires LAI to derive FVC (see Supplementary Section 1). Specifically, we used MOD15A2H LAI data to calculate FVC and validated the results against field-measured FVC, which showed high accuracy (Supplementary Fig. 2b).

We further compared annual FVC estimates derived from SMA and the classical gap fraction method across different grassland types and for all Chinese grasslands. The two methods were strongly consistent, with a correlation coefficient of 0.98 ($P < 0.05$, Fig. R4), and the modeled FVC for each grassland type also showed significant correlations ($P < 0.05$).

These results confirm the robustness of our FVC estimates. In the revised manuscript, we have clarified the rationale for recalculating FVC using MOD15A2H LAI data and the classical gap fraction method, as now stated in lines 155–158 of the main text.

Fig. R4. Comparison of FVC estimates for eight grasslands from 2001 to 2022 derived using the SMA and the classical gap fraction method.

5. The collection of pure pixels was made for the green pixels, and the soil pixels were derived from them? It is difficult to imagine pure pixels for a 500 m spatial resolution.

Response: We agree that identifying pure vegetation and soil endmembers from coarse-resolution remote sensing images is challenging due to mixed-pixel effects. Previous studies have typically defined endmembers using constant values (Li et al., 2014) or percentile-based thresholds (e.g., the 5th percentile for $NDVI_{soil}$ and the 75th percentile for $NDVI_{veg}$) (Wittich and Hansing, 1995; Zeng et al., 2000), largely because of the lack of direct ground-truth observations.

In this study, we applied an iterative optimization method to refine endmember values for different grassland types, aiming to achieve the best fit between modeled FVC and field survey data. The optimized endmembers (Table R1) produced compelling accuracy (Supplementary Fig. 2a). To further evaluate the robustness of SMA-derived FVC, we also applied the classical gap fraction method, which avoids endmember selection. Results again showed strong consistency, with $r = 0.98$ ($P < 0.05$; Fig. R4).

Overall, these analyses provide multiple lines of evidence supporting the robustness of our FVC estimates across the study region. In the revised manuscript, we have clarified this point in the main text (lines 386–391), added Table R1 in the Supplementary Material (now Supplementary Table 2), and included the supporting figure (Supplementary Fig. 2).

Table R1. Parameters of SMA and classical gap fraction for FVC calculation for different grassland types across China.

Grassland types	NDVI _{soil}	NDVI _{veg}	G (0, θ ₁)
Temperate meadow-steppe (TMS)	0.02	0.895	1.00
Temperate steppe (TS)	0.01	0.88	0.42
Temperate desert-steppe (TDS)	0.145	0.515	1.00
Alpine steppe (AS)	0.01	0.5	1.00
Alpine desert-steppe (ADS)	0.01	0.5	0.63
Lowland meadow (LM)	0.13	0.875	0.67
Temperate montane meadow (TMM)	0.01	0.83	1.00
Alpine meadow (AM)	0.01	0.715	1.00

References:

1. Li, Fei, et al. Improving estimates of grassland fractional vegetation cover based on a pixel dichotomy model: A case study in Inner Mongolia, China. *Remote Sensing* 6.6 (2014): 4705-4722.
2. Wittich, K. P., and O. Hansing. Area-averaged vegetative cover fraction estimated from satellite data. *International Journal of Biometeorology* 38.4 (1995): 209-215.
3. Zeng, Xubin, et al. Derivation and evaluation of global 1-km fractional vegetation cover data for land modeling. *Journal of Applied Meteorology* 39.6 (2000): 826-839.

6. It is not clear why allocation of light varied or have an impact on CH under fertilizing effect of rising CO₂.

Response: We agree. In the revision, we have re-evaluated the effects of environmental changes and grazing pressure on CH and FVC. Instead of relying on aggregated annual changes (2001–2022) for statistical analysis, which provided only 22 samples for model fitting, we performed pixel-scale trend calculations. This yielded much larger sample sizes of 4.7 million and 6.3 million pixels for areas with increasing and decreasing CH, respectively. We then applied piecewise SEMs to quantify the independent contributions of environmental and grazing drivers to changes in CH and FVC, and subsequently to AGB. Compared with the earlier analysis based on limited annual samples, this approach more comprehensively captures the regional impacts of the driving factors.

The results revealed that rising CO₂, precipitation, and radiation generally promoted CH and FVC (Fig. R5). In contrast, rising temperature and grazing emerged as the main drivers of CH decline, alongside reduced radiation limitation. To further explore interactions among drivers, we also conducted analyses using LMMs. These indicated significant antagonistic effects of rising temperature on other factors, particularly amplifying the negative impacts of grazing (Table R2). This finding is consistent with previous plot-scale studies (Maestre et al., 2022; Zhang et al.,

2023), which reported that intensified warming increases the risks of grazing-induced losses in biodiversity and ecosystem function, altering both community composition and plant traits.

We have incorporated these findings into the discussion (lines 240–276) and added the new results as Fig. R5 (numbered as Fig. 5) in the main text, with corresponding materials provided as Table R2 (numbered as Supplementary Table 3).

Fig. R5. Responses of CH and FVC to environmental changes in R_a , P_r , T_a , C_a , and G_p . a, b, Path coefficients for grassland areas with increased CH (a), and decreased CH (b), with the calculation of SEM, where the coefficients are asterisked (*) if statistically significant ($P < 0.05$). The embedded area charts present the overall and median statistics of the 2001–2022 average FVC for these two contrasting regions.

Table R2. Effective coefficients of change rates (Δ) in CH and FVC, resulting from environmental factors and grazing pressure, independently and interactively, across areas with increased and decreased CH from 2001 to 2022. The relationships among environmental factors, grazing pressure, CH, and FVC were calculated using linear mixed effects models (LMMs), where the environmental factors and grazing pressure were normalized using the z-score method by subtracting the mean of each term and dividing by the standard deviation for cross-comparison of the independent and interactive effects. A higher absolute value of independent and interactive effects indicates a stronger effect on promoting or diminishing CH and FVC changes. The independent and interactive effects are asterisked (*) if statistically significant ($P < 0.05$).

Driving factors	Areas with increased CH		Areas with decreased CH	
	Effects on Δ CH (+)	Effects on Δ FVC (+)	Effects on Δ CH (-)	Effects on Δ FVC (+)

ΔT_a	-0.38*	-0.19	-0.59*	-0.45*
ΔG_p	-0.19	-0.16	-0.43*	-0.37*
ΔP_r	0.50*	0.27*	0.18	0.20*
ΔR_a	0.58*	0.18	0.66*	0.24*
ΔC_a	0.26*	0.20*	0.29*	0.27*
$\Delta T_a \times \Delta G_p$	-0.35*	-0.14	-0.67*	-0.29*
$\Delta T_a \times \Delta P_r$	-0.26*	-0.20*	-0.06	-0.04
$\Delta T_a \times \Delta R_a$	-0.03	-0.13	-0.05	-0.01
$\Delta T_a \times \Delta C_a$	-0.90*	-0.21*	-0.77*	-0.70*
$\Delta G_p \times \Delta P_r$	0.04	-0.09	0.06	0.26*
$\Delta G_p \times \Delta R_a$	-0.09	-0.06	-0.18	-0.41*
$\Delta G_p \times \Delta C_a$	0.09	-0.26*	-0.17	-0.29*
$\Delta P_r \times \Delta R_a$	0.20	0.15	0.03	-0.02
$\Delta P_r \times \Delta C_a$	0.03	0.15	0.38*	-0.33*
$\Delta R_a \times \Delta C_a$	0.30*	0.15	0.56*	0.13

References:

1. Maestre, Fernando T., et al. Grazing and ecosystem service delivery in global drylands. *Science* 378.6622 (2022): 915-920.
2. Zhang, Minna, et al. Experimental impacts of grazing on grassland biodiversity and function are explained by aridity. *Nature Communications* 14.1 (2023): 5040.

7. It would be important to add a comment on the effect of diminishing CH on the diversity of grass species.

Response: Agreed. Exploring the relationship between CH changes and plant diversity is indeed important. Unfortunately, long-term in situ observations of plant diversity are rarely available. To address this, we analyzed CH changes across different levels of plant diversity, which is directly relevant to the focus of this study. Plant diversity was represented by surveyed species richness and GEDI-derived FHD (Bu and Xiao, 2025).

Our results showed that CH trends were significantly and positively correlated with species richness and FHD ($r = 0.87$ and 0.94) (Fig. R6). Moreover, CH exhibited a declining trend once these diversity indicators dropped below a certain threshold. Previous studies have demonstrated that plant diversity is strongly linked to ecosystem stability, as well as resistance and resilience to environmental change (Yu et al., 2025; Isbell et al., 2015). Our findings further corroborate that CH dynamics are broadly and strongly associated with variations in grassland species diversity, underscoring the importance of biodiversity conservation.

We sincerely thank the reviewer for this constructive comment, which has substantially

improved our study. In the revision, we have added a new section on the relationship between plant diversity and CH changes (lines 284–293 in the main text), together with a new figure (Fig. R6, now Fig. 6 in the main text).

Fig. R6. Correlation between 2001–2022 CH trends and plant diversity across Chinese grasslands. a, Species richness. b, GEDI-derived FHD.

References:

1. Bu, Jingyi, and Jingfeng Xiao. Upscaling eddy covariance measurements of carbon and water fluxes to the continental scale by incorporating GEDI-derived canopy structural complexity metrics. *Remote Sensing of Environment* 329 (2025): 114930.
2. Yu, Qiang, et al. Contrasting drought sensitivity of Eurasian and North American grasslands. *Nature* 639.8053 (2025): 114-118.
3. Isbell, Forest, et al. Biodiversity increases the resistance of ecosystem productivity to climate extremes. *Nature* 526.7574 (2015): 574-577.

Reviewer #2 (Remarks to the Author):

The study reveals China's grassland canopy height (CH) is declining, affecting ecosystem function and food security. It highlights vulnerability to climate change and grazing, guiding adaptive management strategies.

The AGB/FVC ratio method for estimating CH in grasslands addresses remote sensing challenges due to low-density point clouds in LiDAR data, integrating machine learning with multi-source data for high-resolution monitoring.

This study uses the AGB/FVC ratio to estimate grassland CH at a national scale, addressing a gap in previous studies. It provides broad-scale evidence of CH decline, offering a dynamic perspective not previously captured in plot-scale studies.

The work is unique in its methodology for CH estimation and comprehensive analysis of China's

grasslands over two decades, building on established remote sensing and ecological principles.

1.+ The study's reliance on field data from 2009–2016, which does not span the full analysis period (2001–2022), may introduce bias in trend extrapolation, and the use of 500 m resolution data may miss fine-scale variations in CH. To address these limitations, it is recommended to extend the field dataset by including earlier or more recent measurements or validate findings using independent datasets. Using higher-quality satellite data like Sentinel-2 can provide more detailed information, and carefully adding GEDI LiDAR data can help confirm canopy height measurements, even though there are some issues with grasslands. A dedicated section or supplementary analysis should be included to quantify the impacts of these temporal and spatial constraints.

Response: Thank you for providing such valuable comments. While the field measurements from 2009–2016 are highly unique and rare, we agree that it is not sufficient to extrapolate changes across the entire study period based solely on this middle window. Following the suggestion, we validated the CH declining trend during both the early and late periods using independent datasets: long-term observational station data (2001–2018) and GEDI data (2019–2022), respectively. We also recalculated CH trends for the measurement period itself, thereby verifying the robustness of the decline across the full study duration.

For the early period (2001–2018), observations from six independent long-term monitoring sites, together with modeled vertical biomass (AGB/FVC), consistently indicated a decline (Fig. R1). For the recent period (2019–2022), GEDI-derived CH also showed a decreasing trend, with consistent declines across five grassland types (Fig. R3), though we acknowledge notable uncertainties, as the reviewer pointed out. During the measurement period (2009–2016), CH trends derived from modeled data revealed a consistent decrease of -0.02 cm yr^{-1} , with $\sim 53\%$ of China's grasslands exhibiting decline (Fig. R2).

Although the 500 m dataset may not fully capture fine-scale CH variations, we believe this does not undermine the overarching conclusion of widespread CH decline. Regarding finer-resolution data from Sentinel-2 (2017–present), its use for interannual trend analysis may introduce biases, as multiple years are typically required to aggregate complete annual coverage (Li et al., 2024).

In this revision, we have added a section discussing the multiple lines of evidence for CH decline (main text, lines 209–218), along with new figures: Fig. R1, Fig. R2, and Fig. R3 (now Fig. 4, Supplementary Fig. 7, and Supplementary Fig. 8).

Reference:

1. Li, Huaqiang, et al. A machine learning scheme for estimating fine-resolution grassland aboveground biomass over China with Sentinel-1/2 satellite images. *Remote Sensing of Environment* 311 (2024): 114317.

2.+ The manuscript does not provide a detailed analysis of the uncertainties in CH estimates, especially in places with low FVC, where optical remote sensing can give misleading results. To make this better, the authors should analyze how errors affect the CH estimates that come from the AGB/FVC ratio, taking into account the uncertainties in both the AGB and FVC models. Uncertainty should also be explicitly assessed across FVC gradients, particularly in sparse vegetation regions, and confidence intervals should be reported. Additionally, cross-validation metrics such as RMSE and bias should be expanded across grassland types. A dedicated subsection with visualizations like uncertainty maps is recommended to present these results clearly.

Response: Agreed. Following the suggestion, we evaluated the uncertainties of different methods for estimating AGB, FVC, and CH across FVC gradients, and extended this assessment to the grassland-type scale using R^2 and relative error as evaluation metrics. Given the pronounced FVC gradient among Chinese grassland types (Fig. R7), we ranked the modeling uncertainties of the seven grassland types accordingly.

Overall, the three response indicators showed good consistency, with comparable R^2 values. In particular, AGB simulations that combined climatic factors with satellite-derived metrics performed best, while satellite metrics alone introduced larger biases in AGB (Fig. R8a). FVC modelling showed a lower performance in sparsely vegetated alpine desert steppe (ADS), as the reviewer noted (Fig. R8b). However, the classical gap fraction method partly alleviated these biases, improving both R^2 and relative error for ADS. These results suggest that errors in FVC are most evident in sparsely vegetated areas and may propagate into CH estimation (Fig. R8c). However, ADS accounts for only 5.2% of the total grassland area, mainly on the western Tibetan Plateau, and thus is more likely to affect the magnitude of CH change rates rather than the overall declining trend.

In summary, while optical remote sensing inevitably carries uncertainties in sparsely vegetated areas, the relative temporal changes are expected to remain robust. This conclusion is further supported by multiple lines of evidence, including continuous plot-scale observations and GEDI data (Fig. R1 and Fig. R3). To address this point, we have added a new section and Fig. R8 in the Supplementary Material (now Supplementary Section 3 and Supplementary Fig. 9) and discussed it in the main text (lines 218–222).

Fig. R7. Histogram of FVC gradients across seven grassland types.

Fig. R8. Modeling uncertainties across different grassland types ranked by FVC gradients, ranging from 28% in ADS to 88% in TMM. The left panel shows the histogram of R^2 , and the right panel presents the corresponding relative errors. a, AGB. b, FVC. c, CH.

3.+ The paper's claims about reduced ecosystem resilience, biodiversity loss, and grazing impacts are plausible but lack direct supporting evidence, particularly given the non-significant grazing variable. To make these points stronger, the authors should include or reference data on biodiversity, functional traits, or soil carbon to show how the CH decline is connected to ecosystem damage. More detailed grazing data or sensitivity analyses could clarify grazing's role. The authors are also encouraged to propose actionable management strategies, such as rotational grazing or restoration of tall-grass species—and relate these to relevant Sustainable Development Goals.

Response: We agree. In the revision, we incorporated both plot-scale surveyed species richness and GEDI-derived FHD as indicators of species diversity and functional traits, and quantified their relationships with CH trends. The results showed significant positive correlations, demonstrating that variations in species richness and FHD are strongly associated with changes in CH (Fig. R6).

Previous studies have shown that declines in plant diversity induced by overgrazing reduce ecosystem resistance and resilience to climate change (Isbell et al., 2015; Bu and Xiao, 2025). To better capture the influence of grazing, we replaced the previously used annual livestock statistics with nationally gridded 250-m grazing data over the region. Using SEM and LMM approaches at the pixel scale, we examined both the independent and interactive effects of grazing pressure. The results revealed that intensive grazing consistently exerted negative effects on CH and FVC, with particularly strong impacts in areas where CH is declining (Fig. R5). Furthermore, we found that rising temperature amplified the negative influence of grazing (Table R2).

Taken together, these findings provide direct evidence of the ecological risks associated with CH decline, driven not only by grazing pressure but also by climate warming and concurrent changes in species diversity. To mitigate these adverse effects, we propose several actionable management strategies, including rotational and rest grazing combined with revegetation to sustain plant diversity (lines 312–314). While soil carbon is indeed an important topic, given current data limitations and the focus of this study on grassland CH decline (dwarfing), we did not extend our analysis to this perspective at this time.

In summary, the revision re-assesses the impacts of environmental changes and grazing on grassland CH (lines 240–276), and further extends the analysis to associations with species and structural diversity (lines 284–293). These revisions are supported by two new figures and a table: Fig. R5 and Fig. R6 in the main text (numbered as Fig. 5 and Fig. 6), and Table R2 in Supplementary Material (numbered as Supplementary Table 3).

References:

1. Isbell, Forest, et al. Biodiversity increases the resistance of ecosystem productivity to climate extremes. *Nature* 526.7574 (2015): 574-577.

2. Bu, Jingyi, and Jingfeng Xiao. Upscaling eddy covariance measurements of carbon and water fluxes to the continental scale by incorporating GEDI-derived canopy structural complexity metrics. *Remote Sensing of Environment* 329 (2025): 114930.

4.+ The current explanation of asymmetric biomass allocation in the study is overly simplified, emphasizing light optimization without addressing underlying physiological or ecological mechanisms. To deepen the analysis, the authors should integrate data or models on traits like leaf area index (LAI), root biomass, or photosynthetic efficiency to explain growth patterns, and consider interaction effects among drivers such as CO₂ and precipitation.

Response: Thank you for these insightful suggestions. To better clarify the mechanisms underlying asymmetric biomass allocation, we employed SEM and LMM approaches to examine the independent and interactive effects of environmental changes and grazing activity on CH, FVC, and AGB. For this analysis, we re-collected gridded CO₂ and grazing data covering the entire region and conducted pixel-scale assessments separately for areas with increasing and decreasing CH, resulting from sample sizes of 4.7 and 6.3 million, respectively. This substantially enhanced both the stability and comprehensiveness of our results over the region.

The findings indicate that, in addition to the previously noted role of reduced radiation, rising temperature and grazing pressure emerged as the primary drivers of CH decline, with their combined effects further intensifying adverse impacts (Fig. R5, Table R2). In contrast, increased precipitation and elevated CO₂ concentrations had significant positive effects on CH and FVC. These results align with earlier reports that while CO₂ fertilization promotes vegetation greening (Ainsworth and Long, 2005), global warming can amplify the negative impacts of grazing on grasslands (Maestre et al., 2022; Zhang et al., 2023). By comparison, changes in horizontal allocation (FVC) were not as strong as vertical allocation (CH), likely because vegetation tends to expand horizontally under CO₂ fertilization and increased precipitation, whereas vertical growth requires both sufficient moisture and fertile soils but is also constrained by grazing intensity (Falster and Westoby, 2003).

We acknowledge the importance of LAI as a growth indicator. Although LAI was not directly used in this study, the FVC derived from MODIS LAI via the classical gap fraction method showed strong consistency with our SMA-derived FVC (Fig. R4), supporting the robustness of our results. Since the main objective of this work is to report CH decline, we did not extend the analysis to root biomass or photosynthetic efficiency, given both the difficulty of obtaining interannual measurements and the need to maintain a focused narrative.

We sincerely thank the reviewer for helping us refine the mechanistic explanation of CH decline. In the revision, we have substantially improved the discussion of the independent and interactive effects of environmental changes and grazing practices on CH, FVC, and AGB (lines 240–276), supported by the addition of Fig. R5 in the main text (numbered as Fig. 5), and Table R2 in Supplementary Material (i.e., Supplementary Table 3).

References:

1. Ainsworth, Elizabeth A., and Stephen P. Long. What have we learned from 15 years of free-air CO₂ enrichment (FACE)? A meta-analytic review of the responses of photosynthesis, canopy properties and plant production to rising CO₂. *New phytologist* 165.2 (2005): 351-372.
2. Maestre, Fernando T., et al. Grazing and ecosystem service delivery in global drylands. *Science* 378.6622 (2022): 915-920.
3. Zhang, Minna, et al. Experimental impacts of grazing on grassland biodiversity and function are explained by aridity. *Nature Communications* 14.1 (2023): 5040.
4. Falster, Daniel S., and Mark Westoby. Plant height and evolutionary games. *Trends in ecology & evolution* 18.7 (2003): 337-343.

5.+ The paper should improve its global relevance by integrating its findings on China's declining grassland canopy height with global trends. It should compare China's CH decline with other grassland ecosystems, using references like Sloat et al. (2018) and Díaz et al. (2007). Also, highlight unique factors like intensive grazing or climate variability.

Response: In the revision, we compared our findings with several global studies to strengthen their worldwide relevance. Using SEMs, we examined the effects of intensive grazing and climate variability, and highlighted that rising temperature and grazing pressure substantially reduce grassland CH, with amplified effects identified through pairwise LMM analysis (Fig. R5, Table R2).

Our results are consistent with reports elsewhere. For instance, a global meta-analysis of 197 studies by Díaz et al. (2007) showed a widespread shift in grassland species from tall to short forms under grazing pressure. Other studies demonstrated that sustained warming limits biomass accumulation and leads to vegetation dwarfing in global grasslands (Arft et al., 1999; Li, 2018). More recently, Maestre et al. (2022) and Zhang et al. (2023) reported that warming amplifies the negative effects of grazing across the world's native grasslands, driving declines in biodiversity and ecosystem function (Fig. R6). As stated in the main text, our study provides the first dynamic, real-time evidence of CH decline and the adverse impacts of climate warming and grazing at a broad ecosystem scale.

In the revision, we have incorporated these global perspectives and citations, expanding the discussion on CH decline (lines 224–234), driving mechanisms (lines 240–276), and plant diversity (lines 284–293).

References:

1. Díaz, Sandra, et al. Plant trait responses to grazing—a global synthesis. *Global change biology* 13.2 (2007): 313-341.
2. Arft, A. M., et al. Responses of tundra plants to experimental warming: meta-analysis of the

international tundra experiment. *Ecological monographs* 69.4 (1999): 491-511.

3. Li, Dan. Grassland Plants under Warming and Precipitation Change. *Plant Growth and Regulation: Alterations to Sustain Unfavorable Conditions* (2018): 1.
4. Maestre, Fernando T., et al. Grazing and ecosystem service delivery in global drylands. *Science* 378.6622 (2022): 915-920.
5. Zhang, Minna, et al. Experimental impacts of grazing on grassland biodiversity and function are explained by aridity. *Nature Communications* 14.1 (2023): 5040.